# Job satisfaction mediates the effect of self-efficacy on work engagement among physical education teachers in economically disadvantaged areas

Hongping Zhou[1], Shi Qi Xu[1]*, Dong-Hwa Chung[2], De Xin Dang[3,4]*

**1** Institute of Physical Education, Hubei University of Arts and Science, Xiangyang, China, **2** Department of Sports Convergence Technology, Sangmyung University, Seoul, South Korea, **3** Department of Endocrinology, Xiangyang Central Hospital, Affiliated Hospital of Hubei University of Arts and Science, Hubei University of Arts and Science, Xiangyang, China, **4** The First Affiliated Hospital of Guangzhou Medical University, Guangzhou Medical University, Guangzhou, China

* ddx2533387852@gmail.com (DXD); xsq941214@naver.com (SQX)

## Abstract

The unequal distribution of educational resources across regions with varying levels of socioeconomic development remains a global issue. Teachers in economically disadvantaged areas often exhibit lower levels of work engagement due to constraints in resources and limited opportunities for professional growth. This issue is particularly pronounced among physical education (PE) teachers, as PE is frequently regarded as a marginal subject. As a result, PE teachers receive less recognition from administrators and parents, encounter greater professional challenges, and experience diminished work engagement. Enhancing self-efficacy and job satisfaction has been identified as a critical strategy for improving work engagement. However, these relationships among PE teachers in underdeveloped regions remain insufficiently explored. To address this gap, this survey collected 472 questionnaire responses from rural primary school teachers, using a 5-point Likert scale survey. A path analysis model was employed to examine the direct and indirect effects of self-efficacy and job satisfaction on work engagement. The findings reveal that self-efficacy exerts an indirect influence on work engagement, with job satisfaction serving as a key mediating factor. These results suggest that fostering self-efficacy among PE teachers in economically disadvantaged areas can enhance job satisfaction, thereby leading to increased work engagement.

## Introduction

The imbalance in educational resources between regions of varying development levels is a global issue, especially in developing countries and regions [1]. Teachers in economically disadvantaged areas often face a lack of educational resources and professional development opportunities, which negatively impacts their quality of life and leads to negative and perfunctory attitudes [2]. This is particularly evident among physical education (PE) teachers, due to the marginal status of their subject, struggle to develop a strong professional identity,

**Data availability statement:** All data used for this survey are available at figshare at https://doi.org/10.6084/m9.figshare.28365692.v1

**Funding:** This survey was supported by a grant from the National Social Science Fund of China (Research on the Occupational Ecological Dilemma and Collaborative Governance of Rural Physical Education Teachers in the New Era; agreement number 21BTY101).

**Competing interests:** All the authors declare that they have no conflict of interest.

enthusiasm, and investment [3,4]. Consequently, enhancing the work engagement of PE teachers in economically disadvantaged areas has become a focal point for many scholars [5].

Research has shown that enhancing self-efficacy and job satisfaction can significantly improve work engagement. Self-efficacy refers to teachers' confidence in their teaching abilities [6], while job satisfaction reflects the fulfillment derived from their work [7]. Both factors are positively associated with work engagement, fostering a more motivated and productive teaching environment [8–13].

Despite these findings, there remains a lack of research exploring the specific roles that self-efficacy and job satisfaction play in enhancing work engagement among PE teachers in economically disadvantaged areas. Improving the work engagement of PE teachers in economically disadvantaged areas will have significant implications for the development of PE education in these areas. Therefore, this survey aims to understand the impact of self-efficacy and job satisfaction on work engagement among PE teachers in economically disadvantaged areas, offering new strategies for the development of PE education in these areas.

## Literature review and research hypothesis

### Self-efficacy

The theory of self-efficacy was first proposed by Bandura in 1982, emphasizing that motivation and behavior are influenced not only by external factors but also by individuals' confidence [14]. In educational settings, self-efficacy refers to teachers' confidence in successfully fulfilling their educational roles and overcoming challenges [15,16]. It is essential for achieving satisfactory educational outcomes [17,18].

Individuals with high self-efficacy proactively seek challenging roles, invest more time and effort to achieve their goals, and persevere even in the face of setbacks [19]. In educational environments, teachers with high self-efficacy demonstrate better mental health, experience lower levels of burnout and fatigue [20,21], collaborate effectively with colleagues to achieve common educational goals [22], and, importantly, report higher job satisfaction [23]. Ultimately, this leads to improved student academic performance [6,24].

Self-efficacy consists of three key dimensions: instructional strategies, classroom management, and student engagement [25].

Instructional strategies refer to teachers' confidence in selecting and implementing effective teaching methods to achieve educational objectives, impart knowledge, and foster student understanding. These strategies involve creating engaging and interactive classroom environments that stimulate student interest, cater to individual learning needs, and facilitate deeper comprehension [26]. Effective instructional strategies are closely linked to student academic performance, as interactive and student-centered teaching approaches enhance engagement and adaptability to diverse educational settings [27,28].

Classroom management is critical for the effective delivery of educational activities [25,29]. Teachers' confidence in their ability to establish and enforce classroom rules, manage student behavior, and create a conducive learning environment helps maintain focus on educational content [30]. Strong classroom management also increases student engagement [31], reduces teacher stress and fatigue, and enhances job satisfaction [32,33].

Student engagement reflects teachers' ability and confidence in fostering and maintaining student involvement in learning [25]. By designing engaging lesson content, implementing interactive teaching methods, and cultivating positive teacher-student relationships, teachers can enhance student motivation and promote a deeper understanding of the subject matter [34].

## Job satisfaction

Job satisfaction reflects an individual's overall attitude and emotional response towards various aspects of their work, such as salary, promotion opportunities, colleagues, supervision, and work content [35–38]. In educational environments, job satisfaction is a critical factor influencing teachers' attitudes and performance [39]. Decreased job satisfaction among teachers can lead to increased burnout and turnover intentions, which directly impact students' academic performance [40,41]. Moreover, it can contribute to psychological issues such as depression, anxiety, and loneliness among teachers [42–46]. Measures aimed at enhancing teachers' job satisfaction can optimize their educational performance [47,48], help establish effective educational environments, improve educational quality [44], foster collaborative relationships among colleagues [43], and ultimately enhance students' academic performance and learning experience [41].

Teachers' job satisfaction is influenced by various factors, including extrinsic rewards from teaching (such as job recognition and social status), intrinsic rewards related to course teaching, and school-based factors (such as work environment and welfare benefits) [49,50].

Course teaching involves the time, effort, and resources invested in designing, implementing, and evaluating educational activities. Higher satisfaction with course teaching indicates that teachers are content with the preparation and implementation processes. Teachers who are satisfied with course teaching can design appropriate educational content that aligns with educational objectives and student needs, structure effective classroom environments, facilitate organized educational delivery, and stimulate student interest and active learning [34,51].

Welfare treatment encompasses teachers' overall economic, security, and career development benefits, including salary, insurance, retirement benefits, paid vacation, professional training, and promotion opportunities [52]. These benefits directly affect teachers' economic stability, professional security, and career advancement. Teachers satisfied with their welfare benefits typically demonstrate higher efficacy in classroom management, which correlates with improved academic performance among students [53]. Conversely, inadequate welfare can impair teachers' educational performance [54].

Work environment refers to teachers' perceptions and evaluations of school culture, educational resources, and administrative conditions [55]. A supportive work environment provides necessary resources and support, enhances teachers' sense of belonging and professional identity [40,56], fosters respect and recognition for their work, and motivates continuous improvement in professional skills and educational abilities [57].

Job recognition refers to how much teachers feel valued, respected, and supported by school leaders and colleagues. This includes leaders' appreciation for teachers' efforts, collegial relationships, and recognition of professional competence. Feeling supported by school leaders increases teachers' sense of pride and accomplishment, positively influencing their performance and job satisfaction [58]. Similarly, recognition from colleagues boosts teachers' confidence, fosters a positive work environment, and encourages teachers to excel in their teaching roles [59].

Social status measures the level of respect and recognition teachers receive for their professional contributions from various societal sectors, including parents, government agencies, media, community organizations, and the public. This recognition encompasses appreciation for teachers' professional skills, educational achievements, and their role in society and the community [60]. Studies indicate that positive social status in teaching careers correlates with increased work engagement, educational outcomes, professional self-esteem, and feelings of accomplishment [40,61]. When teachers feel recognized socially, they gain confidence and a sense of purpose in their work. Social status not only enhances teachers' enthusiasm for their

work but also motivates them to invest time and effort into implementing innovative teaching methods to improve educational quality [62].

## Work engagement

Work engagement encompasses commitment to work roles, the pursuit of work objectives, and emotional attachment and identification with one's work. High work engagement reflects wholehearted dedication to work tasks, intense focus, and reduced fatigue, all of which contribute to greater personal happiness [63,64]. It is a persistent cognitive state that is not tied to specific events, individuals, behaviors, or goals [65].

As a positive work attitude, work engagement comprises three dimensions: vigor, dedication, and absorption [65,66].

Vigor refers to viewing one's career as an important goal and pursuing it with abundant energy and psychological resilience, even in the face of challenges [67]. Vigor is not only a positive physiological state that enhances work performance [68] but also increases professional happiness [69,70]. Individuals with high vigor are better able to collaborate with colleagues, overcome challenges collectively, and achieve organizational objectives [71].

Dedication is an emotional state characterized by enthusiasm, wholeheartedness, and perceiving work as both important and challenging [72]. Research indicates that individuals high in dedication typically exhibit high work performance [67], job satisfaction [73], organizational loyalty [74], and teamwork [75]. They invest significant time and effort into their work, derive satisfaction and a sense of accomplishment from it, and their enthusiasm positively influences colleagues, fostering teamwork and improving the overall work atmosphere.

Absorption is a heightened state of focus where individuals become fully engrossed in their work roles, experiencing happiness and entering a flow state where time seems to fly by unnoticed [67,72]. Those with high absorption are deeply immersed in their tasks, maintaining high concentration and productivity, which enhances work efficiency and quality [76]. This absorption leads to a sense of accomplishment and satisfaction [77].

## Self-efficacy and work engagement

Self-efficacy for teachers refers to their confidence in achieving positive educational outcomes within their own teaching environments [6]. Teachers who trust in their educational abilities and influence are better equipped to handle various roles and challenges in the educational process. This belief motivates them to invest time and effort into improving their teaching methods, actively participating in educational activities, and persistently enhancing their professional skills and educational performance, ultimately increasing their work engagement. Therefore, self-efficacy is a crucial prerequisite for work engagement [78].

Teachers with high self-efficacy are more likely to implement innovative educational practices, possess better classroom management skills, and engage more effectively in professional development [79,80]. They maintain a positive attitude when facing pressure and challenges in education, devising effective strategies to manage stress and minimize its negative impact on their work, thus sustaining higher levels of work engagement [8]. Research indicates that teachers with high self-efficacy are more motivated, investing greater emotional and mental energy into their work, which results in higher work engagement [81]. Therefore, higher self-efficacy among teachers contributes to increased work engagement [8–10], creating a more positive learning environment [11].

Thus, we hypothesize that an increase in self-efficacy will lead to greater work engagement among PE teachers in economically disadvantaged areas (Hypothesis 1). If Hypothesis 1 is supported, it would suggest that self-efficacy directly influences work engagement among PE teachers in economically disadvantaged areas.

### Self-efficacy and job satisfaction

Self-efficacy among teachers is closely linked to job satisfaction [82–84]. As discussed earlier, self-efficacy refers to teachers' confidence in their ability to plan, organize, and achieve educational objectives within their teaching contexts. Teachers with higher self-efficacy believe in their educational capabilities, enabling them to adapt to diverse educational environments and meet students' needs. This further fosters a sense of accomplishment and satisfaction, which enhances job satisfaction. Research shows that teachers with high self-efficacy derive greater satisfaction from their educational activities [85,86] and report lower levels of burnout [87].

Teachers with higher self-efficacy feel more in control of their work environment and outcomes, improving their ability to manage stress and minimize its negative effects. In contrast, teachers with low self-efficacy may struggle with classroom management and meeting student demands, which can lead to disruptions and persistent challenges. These difficulties can increase stress and hinder their ability to solve problems effectively [88].

Therefore, we hypothesize that job satisfaction among PE teachers in economically disadvantaged areas will increase with higher levels of self-efficacy (Hypothesis 2a).

### Job satisfaction and work engagement

Job satisfaction also plays a significant role in influencing work engagement [89,90]. Teachers who experience high job satisfaction are more likely to feel enthusiastic about their work, investing greater time and effort into teaching and supporting students [91]. Numerous studies have consistently shown a positive relationship between job satisfaction and work engagement among teachers [92–99]. Teachers with high job satisfaction derive internal motivation and purpose from their work, which drives their engagement [87].

Therefore, we hypothesize that work engagement among PE teachers in economically disadvantaged areas will increase with higher levels of job satisfaction (Hypothesis 2b). If both Hypothesis 2a and 2b are supported, it would suggest that self-efficacy indirectly influences work engagement among PE teachers in these areas, with job satisfaction serving as a crucial mediator (Hypothesis 2).

### Conceptual framework

The aim of this survey is to examine the roles of job satisfaction and self-efficacy in enhancing work engagement among PE teachers in economically disadvantaged areas. Based on the theoretical assumptions outlined, this survey has developed a research model (Fig 1).

Fitting degree of model: Minimum discrepancy of confirmatory factor analysis/degrees of freedom, 2.175; goodness of fit index, 0.969; Adjusted goodness of fit index, 0.950; Root mean standard error of approximation, 0.050; Tucker-Lewis index, 0.956; Normed fit index, 0.941; Incremental fit index, 0.967; Comparative fit index, 0.967; Standardized root mean square residual, 0.036

## Materials and methods

The procedures and protocols employed in this survey were approved by the Ethics Committee of Hubei University of Arts and Science (Xiangyang, China).

### Survey design and questionnaire

The questionnaire used in this survey was administered and web-programmed by Wen Juan Xing (Changsha Ranxing IT Ltd.) and distributed via WeChat and email. It consisted of closed-ended questions divided into four sections (S1 Table).

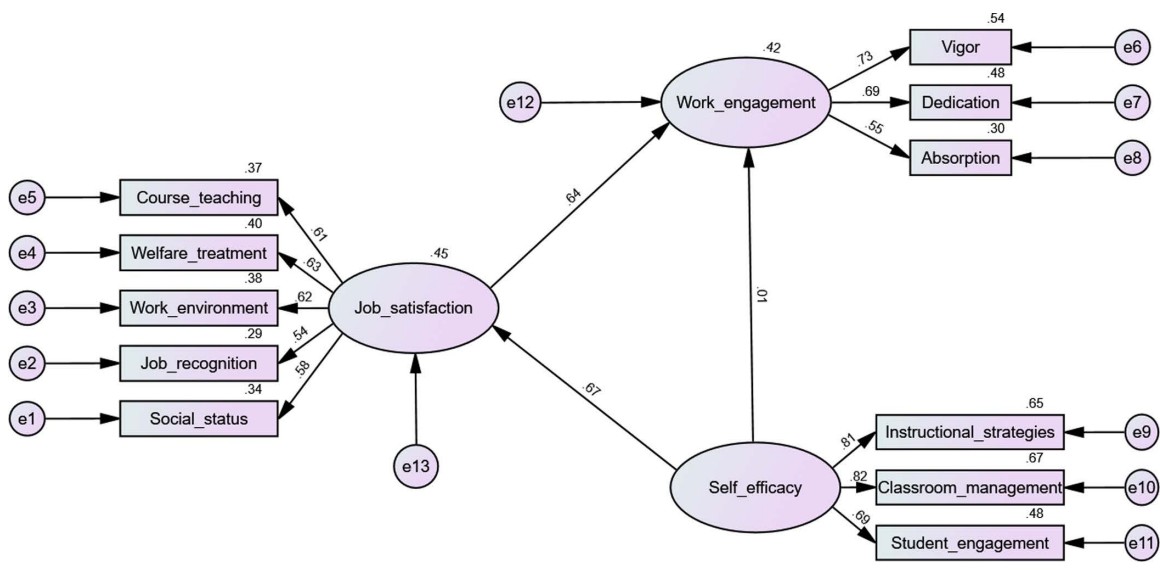

**Fig 1. Path analysis (calculated using AMOS).**

Section 1 comprised 7 questions aimed at collecting socio-demographic information from respondents, including age, gender, educational background, professional title, contract type, weekly teaching hours, and monthly salary.

Section 2 included 24 questions derived from the Teachers' Sense of Efficacy Scale by Tschannen-Moran and Hoy [25] to assess self-efficacy across three variables: instructional strategies (8 questions), classroom management (8 questions), and student engagement (8 questions). Responses were recorded on a 5-point scale.

Section 3 consisted of 19 questions, referencing the questionnaires developed by Chen and Sun [100], Bian [101], and Huang [102] to evaluate job satisfaction across five variables: course teaching (6 questions), welfare treatment (3 questions), work environment (3 questions), job recognition (4 questions), and social status (3 questions). Responses were similarly measured on a 5-point scale.

Section 4 included 17 questions adapted from the Utrecht Work Engagement Scale proposed by Schaufeli et al. [65,72] to assess work engagement across three variables: vigor (6 questions), dedication (5 questions), and absorption (6 questions). Responses were assessed using a 5-point scale.

## Sample size and data collection

The questionnaire was distributed to frontline rural PE teachers from 138 primary schools in Hubei Province, China, through convenience sampling [103]. During the survey period, we visited the above schools and invited the respondents face-to-face. Inclusion criteria required participants to have at least one year of teaching experience and be willing to participate in the survey. Exclusion criteria included dissatisfaction with participation, non-teaching staff, administrators, and incomplete or unreliable responses. Out of 526 responses received, 54 were deemed unusable, leaving 472 valid questionnaires for further analysis.

Before participation, all respondents were informed about the survey's purpose and provided written informed consent. Data collection occurred from 21 March 2022 to 28 May

2023. The sample size was determined using PASS software (version 15.0.5) with a two-sided confidence interval method. A confidence interval width of 0.1, a confidence level of 0.95, an acceptance rate of 50%, and a dropout rate of 10% were used, indicating that at least 428 valid questionnaires were needed. The 472 valid responses exceeded the required sample size [104,105].

### Statistical analysis

Data were analyzed using SPSS (version 26.0). Frequency analysis was employed to examine the general characteristics of respondents. The validity and reliability of the questionnaire were assessed through confirmatory factor analysis and Cronbach's α [106]. Convergent validity was deemed satisfactory for multi-item scales if the average variance extracted value exceeded 0.5 and composite reliability exceeded 0.8 [107].

Research hypotheses were tested using structural equation modeling conducted with AMOS. We followed the two-stage approach proposed by Anderson and Gerbing [108], where the first stage involved confirmatory factor analysis to estimate the model's items, and the second stage examined structural relationships among constructs to test the research hypotheses. Statistical significance was set at $P < 0.05$. Model fit indices, including minimum discrepancy of confirmatory factor analysis/degrees of freedom (CMIN/DF), standardized root mean square residual (SRMR), goodness of fit index (GFI), adjusted goodness of fit index (AGFI), normed fit index (NFI), incremental fit index (IFI), Tucker-Lewis index (TLI), comparative fit index (CFI), and root mean square error of approximation (RMSEA) were assessed against the following thresholds: $< 3$, $< 0.05$, $> 0.9$, $> 0.9$, $> 0.9$, $> 0.9$, $> 0.9$, and $< 0.08$, respectively, to evaluate model adequacy [109].

## Results

### Demographic characteristics of respondents

A total of 472 respondents participated in this survey, of which 37.92% were male and 62.08% were female. The majority were on full-time contracts (98.52%), with only 1.48% on part-time contracts. Regarding age distribution, 19.49% were aged 20–30, 26.91% were aged 31–40, 38.35% were aged 41–50, and 15.25% were over 51 years old. The highest proportion of respondents held a bachelor's degree (88.35%), followed by those with a college degree (9.32%) and a master's degree (2.33%). In terms of professional title, 56.78% held a medium-level title, 14.62% held a junior-level title, and 28.60% held a senior-level title. For teaching workload, 7.42% reported having fewer than 10 lessons per week, 31.14% had 10–15 lessons per week, 41.10% had 16-20 lessons per week, and 20.34% had more than 21 lessons per week. In terms of salary, 3.39% earned 2000–3000 yuan per month, 31.35% earned 3001–4000 yuan per month, 48.10% earned 4001–5000 yuan per month, and 17.16% earned more than 5000 yuan per month (Fig 2).

### Validity and reliability analysis of the questionnaire

Factor analysis of self-efficacy (S1 Fig), job satisfaction (S2 Fig), and work engagement (S3 Fig) revealed no questions with low factor loading values, confirming the adequacy of the measurement items. The Cronbach's α coefficients for all factors supported the internal consistency of the scales. Additionally, the convergent and discriminant validity statistics showed that the average variance extracted and composite reliability values for all multi-item scales exceeded the thresholds of 0.5 and 0.8, respectively, indicating satisfactory convergent validity for the measurement model.

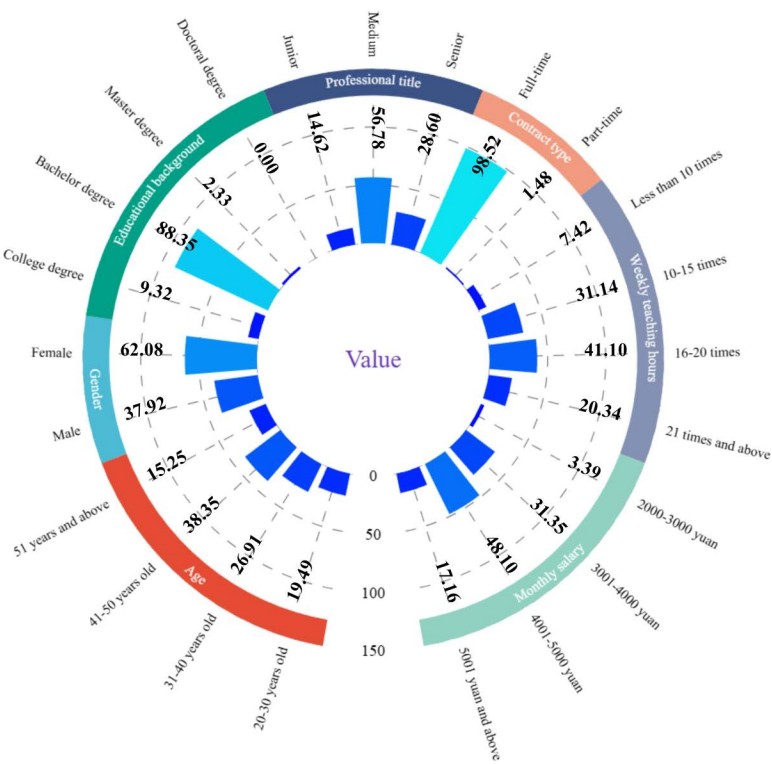

**Fig 2. Demographic characteristics of respondents.**

## Model parameters

The results of the model fit indices confirmed that the proposed measurement model adequately fit the data. The following values were observed: CMIN/DF = 2.175, SRMR = 0.036, GFI = 0.969, AGFI = 0.950, NFI = 0.941, IFI = 0.967, TLI = 0.956, CFI = 0.967, and RMSEA = 0.050 (Fig 3).

## Indirect and total effects

Structural equation modeling indicated that self-efficacy has an indirect effect on work engagement, with job satisfaction serving as a significant mediator in this relationship (Fig 4).

## Hypothesis testing

The hypothesis testing results showed that self-efficacy significantly influenced job satisfaction (self-efficacy → job satisfaction = 0.654, $P < 0.001$), and job satisfaction significantly affected work engagement (job satisfaction → work engagement = 0.390, $P < 0.001$). These findings supported hypotheses 2a and 2b. However, self-efficacy did not directly affect work engagement, leading to the rejection of hypothesis 1 (Fig 1).

## Discussion

This survey explores the role of job satisfaction and self-efficacy in enhancing work engagement among PE teachers in economically disadvantaged areas. Our findings reveal that self-efficacy indirectly influences work engagement, with job satisfaction serving as a crucial mediating factor.

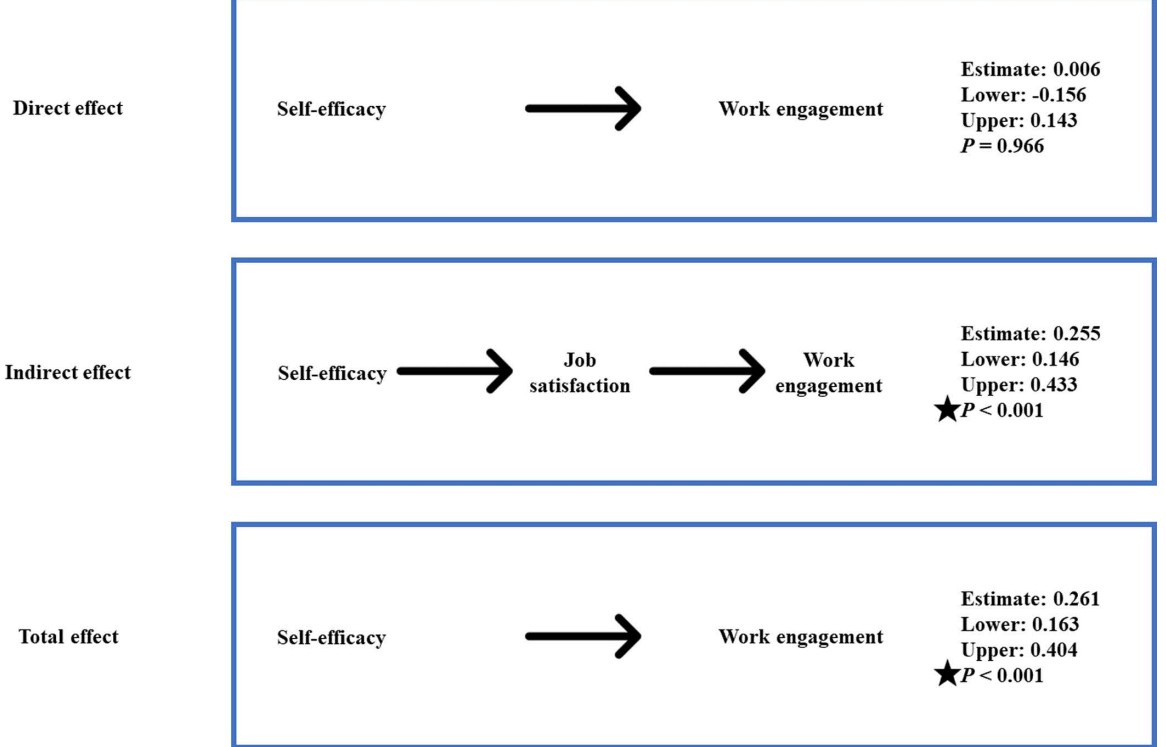

**Fig 3. Results of structural equation modeling.**

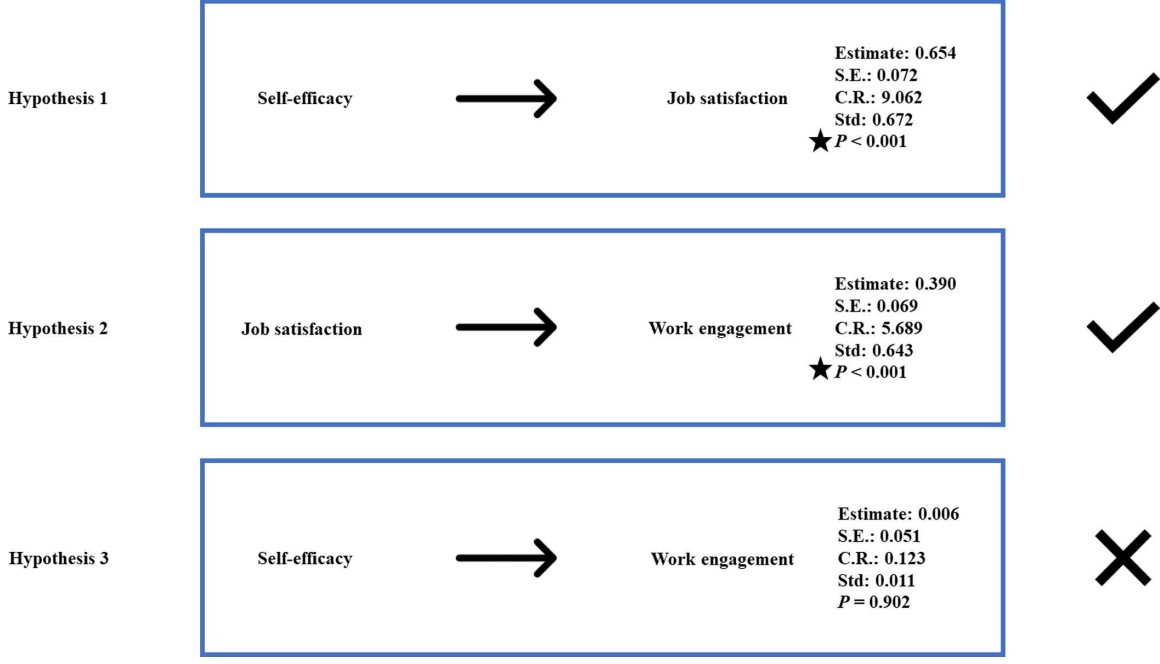

**Fig 4. Summary of support for hypotheses.**

Our results support the notion that self-efficacy has a significant impact on job satisfaction among PE teachers in economically disadvantaged areas. Similarly, Ortan et al. [110] found that increasing teachers' self-efficacy enhances job satisfaction, while Türkoglu et al. [111] identified a strong positive correlation between these two factors. Self-efficacy reflects teachers' confidence in overcoming challenges during the educational process [24]. Teachers with high self-efficacy are more likely to approach educational tasks with a problem-solving mindset, perceiving challenges as manageable [25]. Consequently, they are less prone to frustration or disappointment, leading to greater job satisfaction. Therefore, improving self-efficacy among PE teachers in economically disadvantaged areas can enhance their job satisfaction, validating Hypothesis 2a.

Additionally, our findings confirm that job satisfaction significantly predicts work engagement. Higher job satisfaction fosters a positive impact on work engagement [12]. Oubibi et al. [13] also reported a significant positive correlation between job satisfaction and work engagement in educational settings. Teachers who experience greater job satisfaction tend to invest more time and effort in their professional roles [38,112], exhibiting higher levels of key components of work engagement, including vigor, dedication, and absorption [65]. Thus, enhancing job satisfaction of PE teachers in economically disadvantaged areas is crucial for improving their work engagement, validating Hypothesis 2b.

The validation of Hypotheses 2a and 2b underscores that self-efficacy indirectly influences work engagement, with job satisfaction playing a pivotal mediating role. Strengthening self-efficacy among PE teachers in economically disadvantaged areas can boost their job satisfaction, ultimately leading to higher work engagement. Specifically, increased self-efficacy enables PE teachers in economically disadvantaged areas to approach their work with confidence, fostering positive attitudes and behaviors in the classroom. However, our findings suggest that confidence alone, as proposed in Hypothesis 1, may not be sufficient to enhance work engagement. While self-efficacy positively impacts work engagement, job satisfaction is a necessary condition for achieving this outcome. As noted by Fathi and Derakhshan [20], teachers with high self-efficacy maintain a positive emotional outlook towards their work, which enhances job satisfaction and further facilitates work engagement. Bandura [88] previously emphasized that self-efficacy shapes behavior and performance in multiple ways, with job satisfaction being significant. Similarly, Wang et al. [113] highlight the critical role of job satisfaction in fostering work engagement among teachers.

When teachers feel confident in their abilities, they are more likely to derive satisfaction from their professional roles and work environment, exhibiting higher self-efficacy and further enhancing their work engagement. Therefore, job satisfaction serves as a direct expression of teachers' self-efficacy. Confident teachers gain recognition and fulfillment from their efforts, reinforcing their self-efficacy and ultimately increasing their work engagement. Thus, our findings substantiate Hypothesis 2, emphasizing the essential role of job satisfaction in shaping work engagement.

## Conclusion

This survey highlights self-efficacy as a significant indirect predictor of work engagement among PE teachers in economically disadvantaged areas, with job satisfaction serving as a crucial mediating factor. The findings support the notion that enhancing self-efficacy can effectively increase job satisfaction and further improve overall work engagement. Therefore, fostering both self-efficacy and job satisfaction is essential for promoting work engagement among PE teachers in economically disadvantaged areas.

## Limitations and directions for future research

Despite its contributions, this survey has several limitations.

First, the sample was limited to PE teachers in economically disadvantaged areas of Hubei Province, China, which restricts the generalizability of the findings to PE teachers in economically disadvantaged areas worldwide. Future research should include a more diverse and geographically representative sample to enhance external validity.

Second, this survey relied solely on a quantitative questionnaire survey. Incorporating qualitative methods, such as interviews or focus groups, would provide deeper insights into teachers' experiences and help formulate more targeted intervention strategies.

Third, the survey employed a cross-sectional design, which limits the ability to establish causal relationships among self-efficacy, job satisfaction, and work engagement. While structural equation modeling provides valuable insights into these associations, longitudinal studies are needed to examine causal effects over time.

Fourth, this survey focused specifically on self-efficacy, job satisfaction, and work engagement, without considering other influential factors, such as school management, peer support, and family background. These variables may also play a crucial role in shaping teachers' professional experiences and should be explored in future research.

To our knowledge, this is the first survey to examine the role of job satisfaction and self-efficacy in improving work engagement among PE teachers in economically disadvantaged areas. By providing empirical evidence on this relationship, our survey offers a foundation for developing strategies to enhance PE teacher engagement and reduce educational disparities between developed and underdeveloped regions. Future survey should incorporate qualitative approaches and explore additional contextual factors to gain a more comprehensive understanding of the determinants of work engagement among PE teachers in these settings.

## Research contribution

This survey underscores the importance of recognizing PE teachers in economically disadvantaged areas as key stakeholders whose self-efficacy and job satisfaction significantly impact their work engagement. Educational administrators and policymakers should prioritize initiatives that enhance these teachers' self-efficacy, equipping them with the confidence and resilience needed to navigate professional challenges while maintaining job satisfaction. Additionally, fostering a supportive work environment that promotes job satisfaction is crucial for sustaining these teachers' engagement and long-term commitment to their roles. By investing in strategies that strengthen self-efficacy and job satisfaction, educational institutions can cultivate a more engaged and effective teaching workforce, ultimately improving educational outcomes in economically disadvantaged areas.

## Supporting information

**S1 Fig. Factors and reliability analysis of self-efficacy.**
(TIF)

**S2 Fig. Factors and reliability analysis of job satisfaction.**
(TIF)

**S3 Fig. Factors and reliability analysis of work engagement.**
(TIF)

**S1 Table. Compositions of questionnaire.**
(DOCX)

## Author contributions

**Conceptualization:** Hongping Zhou, De Xin Dang.

**Data curation:** Hongping Zhou.

**Formal analysis:** Hongping Zhou, Dong-Hwa Chung.

**Methodology:** Hongping Zhou, Shi Qi Xu, Dong-Hwa Chung, De Xin Dang.

**Project administration:** Hongping Zhou, Shi Qi Xu, Dong-Hwa Chung, De Xin Dang.

**Supervision:** Hongping Zhou, Shi Qi Xu.

**Validation:** Hongping Zhou.

**Writing – original draft:** Hongping Zhou, Shi Qi Xu, De Xin Dang.

**Writing – review & editing:** Hongping Zhou, Shi Qi Xu.

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
