## [Decision Letter · Decision Letter 0]

16 Dec 2024

PONE-D-24-42818Job satisfaction mediates effect of self-efficacy on work engagement for teachers in economically disadvantaged areasPLOS ONE

Dear Dr. Xu,

Thank you for submitting your manuscript to PLOS ONE. After careful consideration, we feel that it has merit but does not fully meet PLOS ONE’s publication criteria as it currently stands. Therefore, we invite you to submit a revised version of the manuscript that addresses the points raised during the review process.

Below, I have outlined the revisions needed:

**A.**
**Title and Scope Clarification**

The title should be revised to reflect that the study is focused on physical education teachers in a specific context. The introduction, literature review, abstract, discussion, and conclusions should consistently emphasize that the findings are limited to physical education teachers and the specific sample used in the study. This will help avoid contradictions regarding the generalizability of the study.

**B.**
**Sample and Methodology**

Sample Size Calculation: The method used for calculating the sample size should be clearly explained to enhance transparency and methodological rigor.Study Scope: There is concern about the limited scope of the study, which focuses solely on physical education teachers, while the manuscript suggests a broader scope. The authors should clarify this limitation throughout the manuscript.Inclusion and Exclusion Criteria: The inclusion and exclusion criteria for the study should be stated clearly in the methodology section to ensure proper understanding of the study's sample.

**C.**
**Data and Results Presentation**

Presentation of Results: The results section should avoid unscientific language (e.g., "investigation" and "educationally") and ensure that terms are precise and adhere to scientific standards.Questionnaire and Data Analysis: The questionnaire could be placed in an appendix rather than as a table within the main text. The validity and reliability of the questionnaire need to be addressed, particularly if it is to be applied in other contexts. A detailed explanation of the factors considered in the analysis should be provided.Standardization of Data: There are discrepancies in the data presented, such as inconsistencies in table formatting and numerical values. The authors should standardize the data and correct errors in presentation.

**D.**
**Introduction and Literature Review**

Tighten Introduction: The introduction is overly long and should be shortened to improve clarity and focus. It should directly address the research gap and the study's objectives.Literature Review: Redundant content between the introduction and literature review should be removed to improve the flow and coherence of the manuscript. The literature review should also integrate findings relevant to the specific context of physical education.

**E.**
**Formatting and Structural Issues**

Formatting: The manuscript should adhere to PLOS ONE guidelines, particularly regarding formatting issues such as authors' affiliations on the title page, indentation in paragraphs, and titles. Proper attention to these details is necessary to ensure the manuscript meets submission standards.Abbreviations: Ensure that all abbreviations are fully spelled out the first time they are used in the manuscript.Figures and Tables: All figures should meet the PLOS ONE guidelines. It is suggested that Tables 6-11 be consolidated into a single table for better clarity. Figures and tables should also be clearly labeled and referenced within the text.

**F.**
**Discussion and Conclusion**

Discussion of Results: The discussion section should not merely repeat results but should interpret the findings in light of existing literature. It should highlight the implications of the findings, including any possible generalizations, and discuss the significance of the results.Conclusions: The conclusions should be grounded in the study's findings and should avoid overstatements. Recommendations should be specific and related to the study context, such as suggestions for improving laboratory conditions or teaching practices in physical education.

**G.**
**Typographical and Grammatical Corrections**

The manuscript contains numerous typographical, grammatical, and small formatting errors. The authors are encouraged to thoroughly revise the manuscript to ensure proper language use and consistency throughout the document.

We look forward to receiving your revised manuscript.

Kind regards,

Mc Rollyn Daquiado Vallespin

Academic Editor

PLOS ONE

Journal Requirements:

[This study was supported by a grant from the National Social Science Fund of China (Research on the Occupational Ecological Dilemma and Collaborative Governance of Rural Physical Education Teachers in the New Era; agreement number 21BTY101)].

3. In this instance it seems there may be acceptable restrictions in place that prevent the public sharing of your minimal data. However, in line with our goal of ensuring long-term data availability to all interested researchers, PLOS’ Data Policy states that authors cannot be the sole named individuals responsible for ensuring data access (http://journals.plos.org/plosone/s/data-availability#loc-acceptable-data-sharing-methods ).

4. Please include captions for your Supporting Information files at the end of your manuscript, and update any in-text citations to match accordingly. Please see our Supporting Information guidelines for more information: http://journals.plos.org/plosone/s/supporting-information .

Reviewers' comments:

Reviewer's Responses to Questions

**Comments to the Author**

1. Is the manuscript technically sound, and do the data support the conclusions?

Reviewer #1: Yes

Reviewer #2: Yes

Reviewer #3: Yes

2. Has the statistical analysis been performed appropriately and rigorously? 

Reviewer #1: Yes

Reviewer #2: Yes

Reviewer #3: Yes

3. Have the authors made all data underlying the findings in their manuscript fully available?

Reviewer #1: Yes

Reviewer #2: Yes

Reviewer #3: Yes

4. Is the manuscript presented in an intelligible fashion and written in standard English?

Reviewer #1: Yes

Reviewer #2: Yes

Reviewer #3: Yes

5. Review Comments to the Author

Reviewer #1: All comments and observations on the work presented by the authors can be found in detail in the attached file. Authors are requested to pay attention to each of the annotations to ensure the quality and clarity of the manuscript.

Reviewer #2: 1. Perhaps the questionnaire should be placed in an appendix rather than a table in the text.

2. Why physical education teachers and not other teachers? The introduction and review of the literature are about teachers and not specifically physical education teachers.

3. The reference to teachers throughout the text contradicts the restricted sample of physical education teachers used in this study. If any, the findings of this study are limited to physical education teachers only, which you acknowledge in the limitations section.

4. Thus, the title should clearly state “… for physical education teachers in…” and the abstract, introduction, review of the literature, discussion, conclusions, and all relevant sections should reflect this fact.

Reviewer #3: Respected Authors,

The overall quality of your scientific research is good. However, there are some unscientific words like investigation and educationally in the results section.

Secondly, your introduction is so long that it becomes boring. Please tighten up your introduction section.

Thirdly, please explain the sample size calculation method used for selecting your study sample.

Fourthly, there are some formatting issues with your manuscript, like the authors' affiliations on the title page, indents in the first lines of each and every paragraph, and even some titles. Please review PLOS ONE guidelines to adjust them accordingly.

Fifthly, include full descriptions when you are citing the abbreviations for the first time.

Sixthly, please ensure all your figures meet the PLOS ONE guidelines.

6. PLOS authors have the option to publish the peer review history of their article (what does this mean? ). If published, this will include your full peer review and any attached files.

**Do you want your identity to be public for this peer review?** For information about this choice, including consent withdrawal, please see our Privacy Policy .

Reviewer #1: No

Reviewer #2: **Yes: ** Dr. Ali M. AL-Asadi

Reviewer #3: **Yes: ** Dr. Saira Akhlaq

---

## [Author Response · Author response to Decision Letter 0]

7 Feb 2025

Dear editor and honored reviewers

We really appreciate you and the reviewers for your comments on our Manuscript: “Job satisfaction mediates the effect of self-efficacy on work engagement among physical education teachers in economically disadvantaged areas” We express our sincere gratitude and thankfulness for your time and precision in reviewing our manuscript. The responses to the comments are as follows. For your kind information, we have carefully dealt with the comments of the reviewers as follows. We hope the revised manuscript meets the standard of publication. Thank you.

A.Title and Scope Clarification

The title should be revised to reflect that the study is focused on physical education teachers in a specific context. The introduction, literature review, abstract, discussion, and conclusions should consistently emphasize that the findings are limited to physical education teachers and the specific sample used in the study. This will help avoid contradictions regarding the generalizability of the study.

Response: Thank you for your valuable comments. Based on the editor’s and reviewers’ feedback, we have revised the title to: Job satisfaction mediates the effect of self-efficacy on work engagement among physical education teachers in economically disadvantaged areas.

Moreover, we have emphasized that this study is limited to physical education teachers in economically disadvantaged areas in the introduction, literature review, abstract, and other relevant sections. The overall manuscript has been revised to eliminate contradictions regarding the study’s generalizability, as you mentioned. Thanks so much.

B.Sample and Methodology

• Sample Size Calculation: The method used for calculating the sample size should be clearly explained to enhance transparency and methodological rigor.

Response: Thank you for your valuable comments. We have declared the method used for calculating the sample size.

“Before participation, all respondents were informed about the survey’s purpose and provided written informed consent. Data collection occurred from 21 March 2022 to 28 May 2023. The sample size was determined using PASS software (version 15.0.5) with a two-sided confidence interval method. A confidence interval width of 0.1, a confidence level of 0.95, an acceptance rate of 50%, and a dropout rate of 10% were used, indicating that at least 428 valid questionnaires were needed. The 472 valid responses exceeded the required sample size [104,105].”

Please refer to Line 224-229. Thanks so much.

• Study Scope: There is concern about the limited scope of the study, which focuses solely on physical education teachers, while the manuscript suggests a broader scope. The authors should clarify this limitation throughout the manuscript.

Response: Thank you for your valuable comments. We have emphasized throughout the manuscript that this study is limited to physical education teachers in economically disadvantaged areas. Additionally, we have acknowledged this limitation. Please refer to Line 329-332.

“First, the sample was limited to PE teachers in economically disadvantaged areas of Hubei Province, China, which restricts the generalizability of the findings to PE teachers in economically disadvantaged areas worldwide. Future research should include a more diverse and geographically representative sample to enhance external validity.”

Please check it. Thanks so much.

• Inclusion and Exclusion Criteria: The inclusion and exclusion criteria for the study should be stated clearly in the methodology section to ensure proper understanding of the study's sample.

Response: Thank you for your valuable comments. We have disclosed the inclusion and exclusion criteria in Lines 220–222.

“Inclusion criteria required participants to have at least one year of teaching experience and be willing to participate in the survey. Exclusion criteria included dissatisfaction with participation, non-teaching staff, administrators, and incomplete or unreliable responses.”

Please check it on the Line 220-222. Thanks so much.

C.Data and Results Presentation

• Presentation of Results: The results section should avoid unscientific language (e.g., "investigation" and "educationally") and ensure that terms are precise and adhere to scientific standards.

Response: Thank you for your valuable comments. We had two native speakers review and revise the manuscript for grammar. The mistakes you mentioned have been corrected. We hope the grammar now meets your expectations. Thanks so much.

• Questionnaire and Data Analysis: The questionnaire could be placed in an appendix rather than as a table within the main text. The validity and reliability of the questionnaire need to be addressed, particularly if it is to be applied in other contexts. A detailed explanation of the factors considered in the analysis should be provided.

Response: Thank you for your valuable comments. We agree with your feedback. The questionnaire has been submitted as a supplementary file. Additionally, the validity and reliability analysis of the questionnaire is provided in Lines 260–266.

“Factor analysis of self-efficacy (S1 Fig), job satisfaction (S2 Fig), and work engagement (S3 Fig) revealed no questions with low factor loading values, confirming the adequacy of the measurement items. The Cronbach’s α coefficients for all factors supported the internal consistency of the scales. Additionally, the convergent and discriminant validity statistics showed that the average variance extracted and composite reliability values for all multi-item scales exceeded the thresholds of 0.5 and 0.8, respectively, indicating satisfactory convergent validity for the measurement model.”

Please check it. Thanks so much.

For a detailed explanation of the factors considered in the analysis, please see the Statistical Analysis section in Lines 232–235.

“The validity and reliability of the questionnaire were assessed through confirmatory factor analysis and Cronbach’s α [106]. Convergent validity was deemed satisfactory for multi-item scales if the average variance extracted value exceeded 0.5 and composite reliability exceeded 0.8 [107].”

Thanks so much.

• Standardization of Data: There are discrepancies in the data presented, such as inconsistencies in table formatting and numerical values. The authors should standardize the data and correct errors in presentation.

Response: Thank you for your valuable comments. We sincerely apologize for the data-related mistake. The manuscript has been thoroughly checked, and the relevant errors have been corrected. Thanks so much.

D.Introduction and Literature Review

• Tighten Introduction: The introduction is overly long and should be shortened to improve clarity and focus. It should directly address the research gap and the study's objectives.

Response: Thank you for your valuable comments. We have restructured the Introduction section to enhance clarity and focus. Please refer to Lines 32–48. Thanks so much.

• Literature Review: Redundant content between the introduction and literature review should be removed to improve the flow and coherence of the manuscript. The literature review should also integrate findings relevant to the specific context of physical education.

Response: Thank you for your valuable comments. We have removed redundant content between the Introduction and Literature Review sections, as you suggested. Furthermore, we have incorporated relevant findings related to physical education into the literature review. Please check it. Thanks so much.

E.Formatting and Structural Issues

• Formatting: The manuscript should adhere to PLOS ONE guidelines, particularly regarding formatting issues such as authors' affiliations on the title page, indentation in paragraphs, and titles. Proper attention to these details is necessary to ensure the manuscript meets submission standards.

Response: Thank you for your valuable comments. We have reformatted the manuscript to comply with the style guidelines of PLOS ONE. We apologize for the previous formatting errors.

• Abbreviations: Ensure that all abbreviations are fully spelled out the first time they are used in the manuscript.

Response: Thank you for your valuable comments. We have ensured that all abbreviations are fully spelled out upon first mention in the manuscript. Thanks so much.

• Figures and Tables: All figures should meet the PLOS ONE guidelines. It is suggested that Tables 6-11 be consolidated into a single table for better clarity. Figures and tables should also be clearly labeled and referenced within the text.

Response: Thank you for your valuable comments. The citation style has been revised to adhere to the PLOS ONE format.

Additionally, we do not have Tables 6–11. All figures and tables are now clearly labeled and properly referenced in the manuscript

Please check it. Thanks so much.

F.Discussion and Conclusion

• Discussion of Results: The discussion section should not merely repeat results but should interpret the findings in light of existing literature. It should highlight the implications of the findings, including any possible generalizations, and discuss the significance of the results.

Response: Thank you for your valuable comments. We have reorganized the Discussion section. Please check this part. Thanks so much.

• Conclusions: The conclusions should be grounded in the study's findings and should avoid overstatements. Recommendations should be specific and related to the study context, such as suggestions for improving laboratory conditions or teaching practices in physical education.

Response: Thank you for your valuable comments. We have revised the Conclusion section. The suggested contributions have been integrated into the Research Contribution section. Please check it. Thanks so much.

G.Typographical and Grammatical Corrections

The manuscript contains numerous typographical, grammatical, and small formatting errors. The authors are encouraged to thoroughly revise the manuscript to ensure proper language use and consistency throughout the document.

Response: Thank you for your valuable comments. We had two native speakers review the grammar of this manuscript. Additionally, we reformatted the manuscript according to PLOS ONE guidelines. Please check it. Thanks so much.

Journal Requirements:

Response: Thank you for your valuable comments. We have reformatted the manuscript to comply with the style guidelines of PLOS ONE. We apologize for the previous formatting errors.

[This study was supported by a grant from the National Social Science Fund of China (Research on the Occupational Ecological Dilemma and Collaborative Governance of Rural Physical Education Teachers in the New Era; agreement number 21BTY101)].

Response: Thank you for your valuable comments. This statement was incorrect. The correct statement is:

“Hongping Zhou: original draft, conceptualization, methodology, validation, formal analysis, data curation, writing - review & editing, supervision, project administration; De Xin Dang: project administration, methodology, writing - original draft, conceptualization; Shi Qi Xu: writing - original draft, writing - review & editing, supervision, project administration, methodology; Dong-Hwa Chung: project administration, methodology, formal analysis”

We have disclosed this information in the Author Contributions section. Please check it. Thanks so much.

3. In this instance it seems there may be acceptable restrictions in place that prevent the public sharing of your minimal data. However, in line with our goal of ensuring long-term data availability to all interested researchers, PLOS’ Data Policy states that authors cannot be the sole named individuals responsible for ensuring data access (http://journals.plos.org/plosone/s/data-availability#loc-acceptable-data-sharing-methods).

Response: Thank you for your valuable comments. We are happy to share our data. All data used for this survey are available at figshare at https://doi.org/10.6084/m9.figshare.28365692.v1. Please check it. Thanks so much.

Response: Thank you for your valuable comments. We have added captions for the supporting information files at the end of the manuscript. Please refer to Line 654-658. Thanks so much.

5. Review Comments to the Author

Reviewer #1: All comments and observations on the work presented by the authors can be found in detail in the attached file. Authors are requested to pay attention to each of the annotations to ensure the quality and clarity of the manuscript.

Line 26: Search for other keywords, 4 of the 5 listed are included in the title.

Response: Thank you for your valuable comments. We have revised the keywords to: rural teacher, rural education, teacher motivation, educational policy, burnout prevention

Please refer to Line 30. Thanks so much.

Line 35-36: Add citation.

Response: Thank you for your valuable comments. We have added a reference for this sentence. Please refer to Line 37-38. Thanks so much.

“Consequently, enhancing the work engagement of PE teachers in economically disadvantaged areas has become a focal point for many scholars [5].”

Please check it. Thanks again.

Line 48: engagement

Response: Thank you for your valuable comments. The relevant mistake has been corrected. Thanks so much.

Line 71-72: Rewrite, it is equal to lines 38 and 39.

Response: Thank you for your valuable comments. We have removed these sections based on the editor’s feedback. Thanks so much.

Line 78-79: Rewrite, it is equal to line 40.

Response: Thank you for your valuable comments. We have removed these sections based on the editor’s feedback. Thanks so much.

Line 130: “extrinsic” not intrinsic.

Response: Thank you for your valuable comments. The relevant mistake has been corrected. Please check the Line 87. Thanks so much.

Line 161-164: The citation does not correspond to what is written in the paragraph.

Response: Thank you for your valuable comments. We have corrected this ref

---

## [Decision Letter · Decision Letter 1]

3 Mar 2025

Job satisfaction mediates the effect of self-efficacy on work engagement among physical education teachers in economically disadvantaged areas

PONE-D-24-42818R1

Dear Dr. XU,

We’re pleased to inform you that your manuscript has been judged scientifically suitable for publication and will be formally accepted for publication once it meets all outstanding technical requirements.

Kind regards,

Mc Rollyn Daquiado Vallespin

Academic Editor

PLOS ONE

Additional Editor Comments (optional):

Reviewers' comments:

Reviewer's Responses to Questions

**Comments to the Author**

1. If the authors have adequately addressed your comments raised in a previous round of review and you feel that this manuscript is now acceptable for publication, you may indicate that here to bypass the “Comments to the Author” section, enter your conflict of interest statement in the “Confidential to Editor” section, and submit your "Accept" recommendation.

Reviewer #1: All comments have been addressed

Reviewer #2: All comments have been addressed

2. Is the manuscript technically sound, and do the data support the conclusions?

Reviewer #1: Yes

Reviewer #2: Yes

3. Has the statistical analysis been performed appropriately and rigorously? 

Reviewer #1: Yes

Reviewer #2: Yes

4. Have the authors made all data underlying the findings in their manuscript fully available?

Reviewer #1: Yes

Reviewer #2: Yes

5. Is the manuscript presented in an intelligible fashion and written in standard English?

Reviewer #1: Yes

Reviewer #2: Yes

6. Review Comments to the Author

Reviewer #1: The authors have addressed all of the reviewers' comments to improve the quality and clarity of the manuscript, and I therefore give my approval for publication.

Reviewer #2: I thank the authors for addressing all my comments adequately and appropriately. I have no further comments.

7. PLOS authors have the option to publish the peer review history of their article (what does this mean? ). If published, this will include your full peer review and any attached files.

**Do you want your identity to be public for this peer review?** For information about this choice, including consent withdrawal, please see our Privacy Policy .

Reviewer #1: No

Reviewer #2: **Yes: ** Ali M. AL-Asadi

---

## [Editor Report · Acceptance letter]

PONE-D-24-42818R1

PLOS ONE

Dear Dr. Xu,

I'm pleased to inform you that your manuscript has been deemed suitable for publication in PLOS ONE. Congratulations! Your manuscript is now being handed over to our production team.

Kind regards,

on behalf of

Dr. Mc Rollyn Daquiado Vallespin

Academic Editor

PLOS ONE